# Not Only a Formulation: The Effects of Pickering Emulsion on the Entomopathogenic Action of *Metarhizium brunneum*

**DOI:** 10.3390/jof7070499

**Published:** 2021-06-23

**Authors:** Nitsan Birnbaum, Victoria Reingold, Sabina Matveev, Chandrasekhar Kottakota, Michael Davidovitz, Karthik Ananth Mani, Reut Feldbaum, Noga Yaakov, Guy Mechrez, Dana Ment

**Affiliations:** 1Department of Plant Pathology and Weed Research, Agricultural Research Organization (ARO), Volcani Center, Rishon LeZion 7505101, Israel; nitsan.birnbaum@mail.huji.ac.il (N.B.); vickire@gmail.com (V.R.); sabmat94@gmail.com (S.M.); chandrabiotech@gmail.com (C.K.); 2The Robert H. Smith Faculty of Agriculture, Food & Environment the Hebrew University of Jerusalem, Re-hovot 7610001, Israel; karthik@volcani.agri.gov.il; 3Department of Entomology, Nematology and Chemistry Units, Agricultural Research Organization, Volcani Center, Rishon LeZion 7505101, Israel; michaeld@volcani.agri.gov.il; 4Department of Food Sciences, Institute of Postharvest and Food Sciences, Agricultural Research Organization (ARO), Volcani Center, Rishon LeZion 7505101, Israel; reut5489@gmail.com (R.F.); nogay@volcani.agri.gov.il (N.Y.); guyme@volcani.agri.gov.il (G.M.)

**Keywords:** entomopathogenic fungi, defoliator, formulation, fungal disease progression, pickering emulsion, relative humidity, conidia dispersion

## Abstract

Growing global population and environmental concerns necessitate the transition from chemical to eco-friendly pest management. Entomopathogenic fungi (EPF) are rising candidates for this task due to their ease of growing, broad host range and unique disease process, allowing EPF to infect hosts directly through its cuticle. However, EPF’s requirement for high humidity negates their integration into conventional agriculture. To mitigate this problem, we formulated *Metarhizium brunneum* conidia in an oil-in-water Pickering emulsion. Conidia in aqueous and emulsion formulations were sprayed on *Ricinus communis* leaves, and *Spodoptera littoralis* larvae were introduced under low or high humidity. The following were examined: conidial dispersion on leaf, larval mortality, conidial acquisition by larvae, effects on larval growth and feeding, and dynamic of disease progression. Emulsion was found to disperse conidia more efficiently and caused two-fold more adhesion of conidia to host cuticle. Mortality from conidia in emulsion was significantly higher than other treatments reaching 86.5% under high humidity. Emulsion was also found to significantly reduce larval growth and feeding, while conferring faster fungal growth in-host. Results suggest that a Pickering emulsion is able to improve physical interactions between the conidia and their surroundings, while weakening the host through a plethora of mechanisms, increasing the chance of an acute infection.

## 1. Introduction

The world’s population is rapidly growing, with projected global population reaching 9.2 billion people by 2050. This increase, coupled with a rising standard of living, will necessitate an increase of 70% in food production in the next three decades [1,2]. Although the green revolution greatly enhanced food production efficiency, pre-harvest pests still reduce crop yield by a mean of 35% becoming the main hurdle in increasing food production per unit of land. These compounding factors make the time-old problem of pest management, perhaps, more meaningful than ever [1]. Biological control agents are considered at a disadvantage compared to chemical insecticides in terms of replicability, efficiency and speed of kill [3,4,5]. However, an insects’ ability to develop resistance against insecticide [6], along with concerns of environmental impact and public opinion, push research to abridge known disadvantages of microbial control agents. This improvement can be achieved by scanning for more resilient and pathogenic strains, or by improving the formulations used to apply microbial control agents [7,8,9,10].

*Metarhizium* is a genus of soil-borne, anamorphic ascomycete fungi (order: Hypocreales, family: Clavicipitaceae). It may grow endophytically in plant roots, saprophytically on dead matter, or as a pathogen of arthropods. Most entomopathogenic fungi (EPF) do not need to be ingested by the host in order to infect and cause mortality. Instead, conidia attach to the insect integument via non-specific hydrophobic interactions. These interactions are mediated by a rodlet layer on the conidium’s surface, containing hyper-hydrophobic proteins [11,12,13]. These weak bonds are later replaced with more specific protein-mediated interaction [14]. After active attachment, germination on the host integument occurs, which requires high relative humidity (RH) conditions [15]. The forming (or lack of forming) of an appressorium is induced via a set of chemical and tactile cues: (A) physical contact; (B) deficiency in nutrients [16,17] and (C) epicuticular lipids and cuticular composition [13,18]. Once the fungal hyphae penetrated through the cuticle, *Metarhizium* colonize the host’s hemocoel. The fungus may employ various secondary metabolites to overcome the host’s response. After host death, *Metarhizium* switches to saprophytic growth. Under high RH (over 95%), hyphae will break through the cuticle and form aerial conidia. Alternatively, under low RH, the mycelium might sporulate internally, creating chlamydospores [19,20,21]. 

EPF are vulnerable to environmental factors such as ultraviolet (UV) radiation and desiccation. Understanding the influence and damage caused by various abiotic stresses, such as desiccation, can help devise novel formulations and application methods to avoid or remedy these detriments [8,9]. Fungi require high water activity (aw) for growth and proliferation [22]. While the calculated threshold for development at 95.5% RH or aw 0.94 for some strains (24), aw of 0.93 seriously delayed and harmed conidial germination, but did not completely halt it for other strains [15], showing high dependence on high humidity, with strain-specific plasticity. A minimal time of high humidity is required for germination and penetration in order for the infection to be successful [23]. Hence, subjecting inoculated insects to low RH after this critical period does not negatively affect infection rates. These humidity requirements for the onset of infection, traditionally restricted EPF application to humid or protected environments [3]. 

Due to the aforementioned susceptibility to low RH and other environmental factors, formulations are continuously developed and tested to increase shelf-life, improve dispersion during application, improve adhesion to insect cuticle, increase mortality and efficacy and minimize the deleterious effects of sub-optimal conditions [8,9,10]. The hydrophobic nature of Hypocrealean conidia necessitates the use of surfactants [8,10,11] or vegetable- or mineral-derived oils [8,11,24]. Vegetable- or mineral-derived oils are also used as carriers and may disperse conidia far better than water with a surfactant, reducing clumping and improving coverage in the plant after spray application. The low wetting angle between oil and lypophilic surfaces allows for better spread on leaf surface, further improving dispersion, and facilitates in cuticle adhesion [7,8,24]. Furthermore, oils may confer protection of conidia against UV damage [25], heat-stress [26,27], and low RH. Oils were found to retain insecticidal ability of the conidia even at low RH, where aqueous formulations failed to cause mortality [8,20,24,28], although the mechanisms for both UV and desiccation tolerance are not yet fully understood. Lastly, oils are known insecticides, conferring direct damage to insects, and were found to increase fungus-related mortality, regardless of RH [20,29,30,31,32]. In this study we implemented a novel type of formulation for biopesticides based on Pickering emulsion. Pickering emulsion uses nanoparticles as emulsifiers, rather than surfactants. The nanoparticles localize to the oil–water interphase, and may include various inorganic materials, polymers and proteins [33]. Pickering emulsions are considered superior to conventional emulsions in terms of droplet size control, control of rate of release of encapsulated materials, and stability over time and varying environmental conditions. 

In previous trials, *M. brunneum* conidia were individually encapsulated in the oil phase of an oil-in-water Pickering emulsion, stabilized by silica (SiO_2_) particles. The conidia was applied against third instar *Spodoptera littoralis* larvae, and showed elevated conidia-related mortality, while the emulsion itself showed no observable effects on the larvae [34]. The objective of this study was to identify possible mechanisms which enabled the enhanced pathogenicity of *M. brunneum* conidia formulated in a Pickering emulsion. 

## 2. Materials and Methods

### 2.1. Insect Rearing and Fungal Culturing

*Spodoptera littoralis* B. (Cotton leafworm) was chosen as a model foliar pest, as it is a polyphagous lepidopteran foliar pest, affecting many crops in the middle-east. *S. littoralis* insect colony was reared in growth room with 26 ± 2 °C and a photoperiod of 12:12 (light:dark). To synchronize the larvae’s developmental stage, all larvae used were first instar freshly hatched and unfed neonates. 

The fungus used in this study was *Metarhizium brunneum* isolate 7 (Mb7) constitutively expressing GFP reporter gene [18]. Fungal conidia were harvested from a grain-based growing technique [35], as follows: to produce initial inoculum, aseptic conidia from fungal colonies grown on SDA (Difco, France) were harvested. The conidia were suspended in a 0.01% (*v*/*v*) of Triton-X100 (Merck, Germany) solution and adjusted to a concentration of 106 conidia/mL. One ml of conidia suspension was used to inoculate 50 mL of SDB (Difco, France), incubated at 28 °C with agitation (155 RPM) for 3 days, until micro mycelia were visible in the clear fluid (‘fermentation inoculum’). Organic rice was soaked in water for 45 min and dried for 2 h. The soaked rice was divided into roughly 0.5 kg batches and autoclaved. Each batch was inoculated with 35 mL of liquid fermentation inoculum and incubated at 28 °C for 48 h, for the formation of initial mycelium. Bags were then moved to fermentation at room temperature, until the rice grain were completely covered with conidia. 

When the rice-grown mycelia were fully sporulated, the bags were opened in a laminar flow hood to allow a complete drying of the conidia. Conidia were separated from the rice using a set of 2000 and a 600 µm sieves. Harvested conidia were kept in an aseptic 50 mL disposable tube at 4 °C. Before each experiment conidial viability was assessed by overnight germination assay on SDA plates. Conidia were used at >90% germination.

### 2.2. Formulations 

Pickering emulsion was prepared by the Functional Polymers and Nanomaterials laboratory at the agriculture research center at Beit Dagan, as described by Yaakov et al. [34]. In short, the emulsion was prepared by sonicating water and paraffinic petroleum-derived oil at a ratio of 8:2, respectively, with 2% (*w*/*v*) (3-Aminopropyl)triethoxysilane (APTES) silanised SiO_2_ nanoparticles, to allow emulsification. As control we have used distilled water (DW) with 0.01% (*V*/*V*) Triton-X100 (from now on will be referred to as aqueous solution). Triton-X100 was used to lower the water’s surface tension to enable incorporation of the hydrophobic conidia into the aqueous solution. Conidial concentration in both formulations was set to 1 mg of conidia per ml of formulation, the concentration was quantified, using an improved Neubauer chamber (Brand, Germany), and averaged at 4.86 × 10^7^ (SE ± 3.2 × 10^6^) conidia per ml.

### 2.3. Mortality Bioassay of M. Brunneum Infecting Spodoptera Littoralis Larvae

*Ricinus communis* (Castor bean) leaves were collected at the onset of every experiment, and surface sterilized by dipping the leaf for 15 s in a 0.4% (*V*/*V*) bleach solution, followed by two DW washes for 5 s each. Leaves were then sprayed, using a conical nozzle hand sprayer, with 3 mL of the treatment solution per leaf. Treatments included: (1) Control aqueous solution; (2) Aqueous solution with conidia; (3) Control emulsion; (4) Emulsion with conidia. 

Treated leaves were dried completely, cut into squares and embedded in 55 mm Petri plates in 2% agarose. Each plate was populated with 6 freshly hatched, 1st instar larvae. Two RH conditions were conducted for each treatment: (1) plates were sealed with perforated caps and were exposed to the growth chamber’s humidity (from now on referred to as low RH) (Figure 1A) or (2) unperforated caps ~100% RH (from now on referred to as high RH) (Figure 1B). Plates were wrapped with Parafilm (Bemis, Shirley, MA, USA). Plates were then incubated in a controlled growth chamber (Percival-scientific, Perry, IA, USA) at 26 °C and 12:12 photoperiod (light:dark), RH and temperature were recorded throughout the experiment using the LOG32TH thermo-hygro data logger (Dostmann electronic, Wertheim, Germany). Chamber’s RH ranged between 62.1% and 67.4% and averaged at 63.4% for all assays.

Mortality was examined daily over a period of five days; a larva was presumed dead only if touching it with a fine brush did not cause any response. Dead larvae were placed on filter paper (Whatman, Maidstone, UK) impregnated with 500 µL DW and incubated at 28 °C for three days, then examined for mycosis. Mortality assays were repeated three times. Two of the consequent repeats included 10 plates per treatment, while the third repeat included eight plates for a total of 28 plates containing roughly, 170 larvae per treatment.

### 2.4. Conidia Dispersal Assay

Both adaxial and abaxial sides of *Ricinus*
*communis* leaves were sprayed with 3 mL of either aqueous solution or emulsion containing Mb7-GFP conidia. The formulated conidia were allowed to dry completely. Nine mm discs were cut from the leaves and placed on a glass slide, adaxial side facing up. The leaf discs were lightly wetted with water and covered with a cover slip. Each disc was photographed using an Olympus IX81 inverted confocal microscope (Olympus, Shinjuku, Japan) at three random locations, at 20× magnification. Number of Conidia in each field was assessed using the ImageJ software (NIH, Stapleton, USA). The experiment was repeated, with freshly made solutions, three times, two repeats included three discs from each treatment, while one repeat included five discs of each treatment, with a total of 11 leaf discs per treatment. To quantify the geographic dispersion pattern of conidia on the leaf surface the following equation was used: $\left(\mathrm{VMR}=\frac{S^{2}}{\bar{X}}\right)$, where S^2^ is the variance and $\bar{X}$ is the mean based on the conidia count of three fields. VMR (variance-to-mean ratio) is an estimate for geographic dispersion pattern of organisms in ecology, by its deviance from the value of 1, the VMR of a Poisson distribution [36,37]. When VMR = 1, individuals are randomly dispersed in a two-dimensional space. If VMR is significantly lower than 1, individuals are regularly spaced geographically and if VMR significantly exceeds 1, individuals are clustered.

### 2.5. Conidia Adhesion Assay

*Ricinus communis* leaves were sprayed with Mb7-GFP conidia in either aqueous solution or emulsion. The leaves were processed and populated with larvae as described in Section 2.3. Larvae were left to settle and roam the leaves for approximately 24 h at 26 °C and low RH. Larvae were then collected from the leaves and placed in a 90 mm petri plate. Larvae were anesthetized by placing the larvae-containing plates on ice. For assessing conidial acquisition, larvae were placed on a microscopic slide with a drop of water and flattened using a cover slide. Larvae were then photographed using an inverted confocal microscope at 20× magnification and conidia were counted from the images using the ImageJ software. For assessing active conidia adhesion, unbound conidia were detached based on the protocol of Ment et al. [38] as follows: collected larvae were washed in 1 mL of tap water for one minute on a MiniMixer tabletop shaker (Benchmark, Lodi, CA, USA) and processed as above. Passive acquisition experiments included four larvae per treatment for three repeats, and five larvae per treatment for the fourth repeat for a total of 17 larvae per treatment. Active adhesion experiments included eight larvae per treatment in three repeats for a total of 24 larvae per treatment.

### 2.6. Disease Progression via Direct Observation

Larvae were introduced to treated leaves, as described in Section 2.3. Five plates were sacrificed each day over a three-day period. Larvae were fixed to a microscopic slide using a double-sided clear cellulose tape. Larvae were examined using an inverted confocal microscope. Infection was divided in to six statuses (Figure 7): (A) no conidia adhered to the insect cuticle, (B) non-germinated conidia on cuticle, (C) conidial germination and appressoria formation, (D) cuticle penetration, (E) early infection of the hemocoel, (F) high infection of the hemocoel. As this experiment was qualitative, a larva’s infection status was determined by the most advanced fungus found on the larva. The experiment was repeated three times. This experiment was also used to assess sub-lethal effects. For sample sizes for each treatment and time point, see Table 1.

### 2.7. Sub-Lethal Effects on Larvae

#### 2.7.1. Larval Growth Rate

After observing and recording disease progression, glued larvae were taken and photographed using a Nikon SMZ645 stereomicroscope (Nikon, Tokyo, Japan) and a Dinoeye edge eyepiece camera (AnMo Electronics, Taipei, Taiwan) to assess larval length. Millimeter paper (Hadar, Jaffa, Israel) was added for scale. Dead larvae were omitted as their time of death was unknown and, therefore, could not represent growth over the given time period. Furthermore, the dead larvae, while being of roughly the same size regardless of treatment or day post-inoculation (DPI), greatly affected variance and significance between treatments. The experiment was repeated three times, alongside the disease progress experiments. For the total sample size for larval length measurement, see Table 1.

#### 2.7.2. Effects on Larval Feeding

To assess feeding ability by larvae, damaged leaf area was measured. Plates were photographed after larvae were sacrificed with millimeter paper as background for scale (Figure 2). Damaged leaf area was manually assessed using the ImageJ software. Damaged leaf area was recorded in three repeats. Each repeat included five plates, for a total of 15 plates per treatment.

### 2.8. Statistical Analysis

All statistical analyses and calculations were performed using the JMP^®^ software, version Pro-15 (SAS institute Inc., Cary, NC, USA). Graphs were formulated using the Excel spreadsheet software (Microsoft, Redmond, WA, USA). The decision rule was fixed for all tests as α = 0.05. All experiments were repeated at least three times. Repeats were counted as random blocks and were inserted in all analyses as random variables and were analyzed using the restricted maximum likelihood (REML) method.

#### 2.8.1. Parametric Analyses

To analyze fifth day mortality (Section 2.4), larval length and larval feeding area (Section 2.8) a three-way analysis of variance (ANOVA) in a mixed model was employed, with least squares analysis for RH, presence of conidia and formulation as fixed variables in a full factorial model. Multiple comparisons were conducted with Tukey’s honestly significant difference (HSD) method on the triple interaction. VMR and conidia count in dispersal assays (Section 2.5) as well as numbers of acquired and adhered conidia (Section 2.6) were analyzed using a one-way ANOVA in a mixed model, with formulation as a fixed variable and repeats as random variables. Equality of variances was tested for VMR and conidia count in dispersal assays, using Bartlett’s test. As variances were not equal, a Welch’s ANOVA was employed

#### 2.8.2. Transformations and Data Normalization

The number of conidia per microscopic field, observed in the dispersal assay, were log_10_ transformed. The number of conidia acquired or adhered to larvae were square root transformed. Numbers of dead larvae in the fifth day mortality were weighted against sample size at time zero, to account for setup variability, through variance scaling. For all other analyses, studentized residues were calculated and observations deviating from the mean by more than two standard deviations (SDs) were omitted. In the larval length analysis, dead larvae were omitted, as they did not represent larval growth and largely increased variance.

#### 2.8.3. Non-Parametric Analyses

Differences in mycosis rates were examined using a Fisher’s exact test, with formulation as an explaining factor. The results for each RH were analyzed separately. Disease progression was analyzed using a Wilcoxon rank-sum test, with RH, formulation, RH*formulation combination, and the date of repeat as independent variables. Multiple comparisons were performed by an each-pair Wilcoxon test. A Holm–Bonferroni correction was employed to counter family-wise error rate (FWER). 

#### 2.8.4. Synergism via the Bliss Model

Synergism was tested according to Bliss’s model for independent action of two chemical pesticides in a mixture [39]. According to Bliss’s model, in case there is no interaction between two toxins in an insecticide, insects will die of unknown reasons (natural mortality), or of active ingredient A, or of active ingredient B. This was mathematically formulated as follows: P = p_0_ + (1 − p_0_) × p_1_ + (1 − p_0_) × (1 − p_1_) × p_2_. Where P signifies the expected probability of mortality from the given mixture, p_0_ signifies probability of natural mortality, p_1_ signifies the probability of mortality from ingredient A, and p_2_ signifies the probability of mortality from ingredient B. P was calculated independently for high and low RH. p_0_ was the mean percentage of mortality in plates containing control aqueous solution-treated leaves. p_1_ was the mean percentage of mortality in plates containing aqueous solution with conidia-treated leaves. p_2_ was the mean percentage of mortality in plates containing control emulsion-treated leaves. Mean mortality of the mixture was calculated according to the mortality percentage of individual plates in the emulsion with conidia group, with a 95% confidence interval. If the expected P was lower than the lower confidence interval of the mortality from emulsion with conidia, the mixture was considered synergistic.

## 3. Results

### 3.1. Effects of Conidia, Formulation and Relative Humidity on Larval Mortality

Mortality of *S. littoralis* was assessed in 2 humidity levels for a 5-day period. Emulsion treatments caused higher mortality than aqueous solution treatments by an order of magnitude (Figure 3 and Table 2). Conidia in aqueous solution resulted in mortality similar to aqueous solution without conidia (control group). At high RH, conidia in emulsion caused greater mortality than control emulsion throughout the five days of the experiment, while at low RH, mean mortality caused by conidia in emulsion had only surpassed that of control emulsion by the fifth day and only by 8.5% (Figure 3 and Table 2). 

All fixed effect factors, and their interactions, were significant (Table 3). Multiple comparisons using the Tukey-HSD method on the fifth day mortality for the triple interaction, found that mortality was significantly highest in larvae exposed to conidia in emulsion at high RH, followed by control emulsion at high RH, conidia in emulsion and control emulsion at low RH. No significant difference was found between larvae exposed to the various aqueous solution treatments at the two humidity levels (see Figure 3 and Table 2). Control emulsion at low RH was not significantly different from conidia in emulsion at low RH nor from the various aqueous solution treatments. To connect mortality with mycosis, cadavers were incubated in a moist chamber for 3–4 days at 28 °C, as described in Section 2.3. By the fifth day post-inoculation, cadavers of larvae exposed to conidia formulated in emulsion at high RH showed a higher mycosis rate by 39.6% (6 of 11 for conidia in aqueous solution vs. 99 of 130 for conidia in emulsion), although this difference was insignificant (Fisher’s exact test *p*-value = 0.1482). Mycosis rates under low RH ere practically indistinguishable (2 of 4 for conidia in aqueous solution vs. 20 of 39 conidia in emulsion) and were statistically identical (Fisher’s exact test *p*-value = 1.0).

Synergism between conidia and emulsion was assessed using the Bliss model for toxins in a mixture, as explained in Section 2.8.4. The observed mortality was significantly higher than the expected mortality at high RH. The difference between observed and expected mortality at low RH was not significant, although it showed an observed mortality higher by 1.5-fold than predicted mortality (Table 2).

### 3.2. Distribution of Conidia Applied in Emulsion or in an Aqueous Solution

To assess conidia distribution patterns on the leaf, leaf discs were used to assess average number and variance of conidia per microscopic field. A variance-to-mean-ratio $\left(\mathrm{VMR}=\frac{S^{2}}{\bar{X}}\right)$ was calculated to quantify distribution pattern. Although conidia formulated in emulsion showed a less clustered dispersion relative to conidia formulated in aqueous solution by almost three fold, this difference was not significant (values of VMR were 16.82 vs. 48.9, respectively. F = 2.9551, Welch’s *p*-value = 0.1236, Figure 4A). However, the variance was significantly different between the two formulations (Bartlett’s test: F = 19.5792 *p*-value < 0.0001), with the variance of the VMR in aqueous solution being higher (Figure 4A). Although VMR values were not significantly different between formulations, the VMR of the emulsion did not significantly differ from 1 while aqueous solution was significantly higher than 1 (t = 1.7134, *p*-value = 0.0587 and t = 2.4713, *p*-value = 0.0209, respectively). This implies that while conidia in emulsion were randomly distributed, conidia in aqueous solution were clustered. In respect to the mean number of conidia per microscopic field, more conidia were observed in the aqueous solution formulation than in emulsion, this difference was not significant (138.26 vs. 127.31 respectively. F = 0.3765, Welch’s *p*-value = 0.5424, see Figure 4B). As in the case of VMR, variance in the number of conidia was significantly higher in aqueous solution than in emulsion (Bartlett’s test: F = 8.4121 *p*-value = 0.0037, Figure 4B). 

### 3.3. Acquisition and Adhesion of Conidia Formulated in Emulsion or Aqueous Solution to Larvae’s Cuticle

To assess the formulation effect on conidial acquisition and adhesion to larval cuticle, larvae were examined under confocal microscopy. By passive acquisition (unbound conidia), conidia formulated in emulsion were acquired almost two-fold more than conidia formulated in aqueous solution. This difference between formulations was statistically significant (84.65 vs. 46.27 conidia, respectively, F = 4.3518, *p*-value = 0.0463, Figure 5). Furthermore, conidia in aqueous solution exhibited more larvae with large uncountable clusters on their cuticle, showing such clusters in 3 out of 12 larvae observed. Conidia in emulsion exhibited only one larva with uncountable clusters out of 12 larvae observed, further demonstrating the clustering shown in Section 3.2. Active adhesion (bound conidia) behaved similarly, with a two-fold advantage in adhesion for conidia formulated in emulsion than for conidia in aqueous solution (11.43 vs. 5.95 respectively, F = 16.1616, *p*-value = 0.0002, Figure 5). As in passive acquisition, conidia in aqueous solution exhibited more larvae with conidial clusters on their cuticle, with 6 out of 24 larvae observed, vs. 3 out of 24 larvae observed for conidia in emulsion.

### 3.4. Sub-Lethal Effects of Conidia, Formulation and Relative Humidity on S. Littoralis Larvae

#### 3.4.1. Effects of Conidia, Formulation and Relative Humidity on Larval Body Length

Larval development was assessed by measuring larval length. Throughout the three examined days, larvae exposed to aqueous solution treatments were longer than larvae exposed to emulsion treatments, regardless to the presence of conidia in either formulations. Furthermore, larvae exposed to low RH tended to be longer than larvae exposed to high RH in the same treatment. The presence of conidia did not affect body length, with no clear trend throughout the three examined days (Table 4). Formulation and RH were significantly influential on larval body length throughout the three days of experiment. Conidia*formulation interaction was significantly influential throughout the three days of the experiment. RH*formulation interaction was significantly influential on larval length on the first and second DPI, yet not on the third DPI, while RH*conidia interaction was only significantly influential on the third DPI. The triple interaction of RH*conidia*formulation was not significant (Table 5). 

#### 3.4.2. Effects of Conidia, Formulation and Relative Humidity on Leaf Area Damaged by Larvae

Assessing leaf area damage was carried out by photographing leaves before and 3 days after larval introduction, as indicator for sub-lethal effect on feeding. Larvae exposed to aqueous solution treatments caused a larger area of damage than larvae exposed to emulsion. In a similar fashion, larvae exposed to low RH caused greater damage than larvae exposed to high RH, for the same treatment groups. The presence of conidia, however, caused a difference in damaged area only in aqueous solution treatments, with conidia-treated leaves being more damaged than corresponding control leaves (Figure 6). All fixed effect factors were significant, save the RH*conidia and the RH*formulation*conidia interactions (Table 6). 

### 3.5. Effects of Formulation and Relative Humidity on M. brunneum Disease Progression

Disease progression was assessed using confocal microscopy observation of larvae after 1–3 days following introduction to inoculated leaf segments. Data gathered over the different repeats have shown different trends according to the days post-inoculation. On the first DPI, aqueous solution formulated conidia at low RH showed a more advanced disease, with the highest percentage of larvae showing germinated conidia. Emulsion formulated conidia at high RH showed the lowest disease progression, with the highest percentage of larvae with no adhered conidia. Aqueous solution at high RH and emulsion at low RH showed comparable disease progression that were intermediary (Figure 7 and Figure 8A). These groupings were significant by multiple comparisons, though the difference between emulsion at high RH and the intermediary group was nullified by a Holm–Bonferroni correction. All the explaining factors were statistically significant (Table 7). On the second DPI, emulsion treatments had shown higher percentages of post-penetrational disease stages than those of aqueous solution treatments, regardless of the presence of conidia (Figure 8B). Concordantly, formulation caused a significant effect on the results, while RH and the RH-formulation combination were insignificant (Table 7 and Figure 8B). On the third DPI, emulsion formulated conidia at high RH showed a greater disease progression, with a high percentage of larvae showing high infection, while all other treatments were comparable. Concordantly, multiple comparisons found emulsion at high RH to be significantly different than all other treatments (Figure 8C). All explaining factors were significant (Table 7).

## 4. Discussion

This research was initiated by successes of earlier trials and had the main objective of uncovering possible mechanisms that lead to the efficient mortality observed by conidia formulated in an oil-in-water Pickering emulsion. These mechanisms were examined using various methods, focusing on the emulsion’s effects on physical aspects of conidial application, physiological effects on the conidia and physiological effects on the larvae. Uncovering said mechanisms can serve as a foundation to discern general traits required for impending formulations.

### 4.1. Mortality of Larvae Exposed to Conidia in Different Formulations at Optimal and Sub-Optimal Humidity

Due to the complexity of pathogenic systems, the most common comparison of mortality rates would be to our previous study [34]. Although mortality in the present study of larvae exposed to emulsion-formulated conidia and control group, at high humidity, were in accordance with those of Yaakov et al. [34], control emulsion and aqueous solution formulated conidia scored much higher and much lower mortality rates, respectively, than previous trials. These differences can be attributed, mostly, to larval developmental state. While Yaakov et al. [34] examined third instar larvae, this work examined first instar larvae. The difference in developmental stage can explain the elevated susceptibility of larvae to the emulsion treatments observed in this study, especially when considering the anti-feedant effect of the emulsion. In this study, control emulsion showed noticeable killing capacity, though this difference was not significant from the control group under low RH, similar results were reported for peanut oil in migratory locust (*Locusta migratoria*) [20]. The mortality caused by the emulsion can be attributed to the toxicity of the oil, including changes in behavior and disruption of the cuticle and various tissues, as reviewed by Buteler and Stadler [30]. Furthermore, our work reveals an increase in mortality for conidia formulated in emulsion, in comparison to conidia formulated in aqueous solution, regardless of humidity. This oil induced enhancement of mortality was also reported for several orders of insects and mites [31,32,40,41], where these successes were attributed to epicuticle disruption and enhanced adhesion of conidia to the cuticle.

Using the Bliss model for independent action of two toxins in a mixture [39], showed a significant difference between mortality predicted in the model and those observed at high RH, corroborating the notion that the increased mortality was caused by a synergistic effect between the conidia and emulsion. At the low humidity conditions synergism was not demonstrated and emulsion formulated conidia did not cause significantly higher mortality than control emulsion. This suggests a role of high humidity in mortality, as humidity close to saturation may cause death and sub-lethal effects on various insects [42,43]. Although no significant difference was recorded at low RH on the fifth day post-inoculation, an advantage for emulsion-formulated conidia can be seen in the increasing rate of mortality over time (manifested by an increased slope) against the decreasing rate of mortality for control emulsion. A longer assay might have yielded a synergistic effect, as low RH is known to delay germination and mortality [44]. 

### 4.2. Dispersion Pattern of Conidia after Spray Application on Leaves

Aerial conidia of various EPF from the order Hypocreales were found to be extremely hydrophobic, this trait was correlated to conidia’s ability to initially attach to a host cuticle [11,45,46]. Such surface traits, manifested in a preference of conidia to the organic phase in phase separation assays, necessitate formulations that will deal with the hydrophobicity of conidia. This can be done by using surfactants that lower the surface-tension of the water, or by using a hydrophobic fluid such as vegetable or mineral oils [7,8].

Results showed an advantage for the emulsion in distribution pattern in several ways. Inspection of the variance-to-mean ratio showed that aqueous solution had a clustered distribution of conidia, while emulsion showed a random (non-clustered) and more uniform distribution. This can also be exemplified by the higher amount and size of clusters observed on the leaf surface in aqueous solution in comparison to the emulsion. A comparison between the number of conidia observed per microscopic field showed a small, insignificant difference in favor of aqueous solution-formulated conidia. Interestingly, variances were not equal, with the variance of conidia in aqueous solution being significantly higher. These results indicate that emulsion is able to distribute conidia in a more uniform manner than a solution of 0.01% Triton-X100. This is facilitated by two main factors: (A) conidium-formulation interaction and (B) the nature of the emulsion. As mentioned, EPF conidia of many species, *M. brunneum* included, are highly hydrophobic [11,45,46]. Therefore, conidia disperse more readily in oil than in water-based formulations, even with surfactants used to decrease the water’s surface tension [8,24]. This can account for the clustering in aqueous solution and the lack of it in an oil-in-water emulsion. Furthermore, the nature of a Pickering emulsion may further improve distribution, firstly, the size of the droplets can be easily controlled and stays in a relatively narrow spectrum, as mentioned by Yaakov et al. [34], physically restricting the number of conidia that can be encapsulated by each droplet. Secondly, as Pickering emulsions utilize solid nanoparticles, as an alternative to surfactants, causing droplets to behave as rigid colloids, reducing droplet coalescence [34] and with it conidia clustering.

### 4.3. Conidial Acquisition and Adhesion to Larvae from Leaf Surface

Conidia of Hypocrealean EPF attach to a host’s cuticle in a two-stage process. At first, conidia are acquired via a non-specific hydrophobic interaction between the conidium’s outer layer and the host’s hydrophobic epicuticle [11,13]. After the initial nonspecific contact, a specific, protein-mediated, attachment is formed to substitute the weak hydrophobic interaction [14]. According to the results obtained, in both cases, passive acquisition and active adhesion, larvae exposed to emulsion formulated conidia had a fungal load significantly higher than larvae exposed to aqueous solution formulated conidia. Although the experimental design did not allow repeated measures, an acquisition to adhesion ratio can be calculated based on mean conidia counts for an adhesion ratio of 13.5% and 12.96% for emulsion and aqueous solution, respectively. This negligible difference hints that the difference in adhered conidia comes from a difference in the amount of conidia acquired, rather than a change in conidia’s response to the cuticle as a result of formulation.

The higher acquisition could be explained by the better dispersal of conidia in oils and emulsions on the leaf surface [8,24]. This can also explain why fewer larvae were found with conidial clusters on the cuticle in emulsion formulation than in aqueous solution. Another explanation for this phenomenon arises from Burges’ [8] review on EPF formulations, stating that oils spread rapidly on both insect and plant cuticles, transferring conidia to the less-accessible folds of the intersegmental membranes (in the case of an insect cuticle), before absorption of the oil into the cuticle. In this project, oil droplets were observed via bright-field microscopy on leaf segments, but not on the insect cuticle. With respect to these evidences, it is possible that larval movement disrupted the rigid shell of the oil droplets, allowing absorption of the oil and the conidia contained in it, although this theory will need a more rigorous study into the physical interaction between larvae and emulsion to be corroborated or debunked. Another possible explanation might lay with the physical properties of paraffinic oil. Akbar et al. [47] and Lord [48] found that desiccants increase the efficacy of *B. bassiana* by disrupting the insects’ cuticle and allowing for more attachment of conidia. As paraffinic oil disrupts or dissolves the epicuticular layer [30], it is possible that this disruption allows for greater conidial acquisition.

### 4.4. Sub-Lethal Effects of Formulations, Conidia and Environmental Conditions on S. littoralis Larvae

EPF are considered at a disadvantage in comparison to conventional insecticides, as they take longer to kill the host. Although the mortality of pests is a gold standard for insecticidal efficacy, it was suggested that the measure of crop damage should be the main criteria of a pesticide’s efficacy [4,49]. Therefore, we investigated sub-lethal effects of conidia, formulation, and RH on larvae. Paraffinic oil is a known and potent insecticide and acaricide, used before the onset of synthetic pesticides [30]. Although the main hypothesized mode of action was considered to be suffocation through tracheal blockage, modern studies show that paraffinic oils cause death through a plethora of mechanisms [29,30,50]. In this project, we have shown that paraffinic oil possesses a toxicity by itself, as well as synergistically enhancing the lethal action of *M. brunneum*. In respect of sub-lethal effects, emulsion showed a significant and substantial reduction in both larval growth and leaf area eaten. It is well established that paraffinic oil causes a feeding deterrence [29,30,51]. This reduction in feeding also best explains the reduced length of larvae exposed to emulsion treated leaves. Feeding deterrence can also be achieved using botanical extracts, such as azadirachtin, and other extracts of Meliaceae plants, yet their short half-life and the inconsistent reaction of various pest species might hinder their use as adjuvants to EPF [49,52].

Relative humidity is considered a determinant factor for EPF germination, and with it the onset of disease [15,44]. However, exposure of insects to RH higher than 90% is detrimental by itself [42]. Extreme high humidity, as was set in the high RH treatments (~100%), causes prolonged larval and pupal periods in *S. littoralis* and *S. frugiperda* [53,54,55]. Furthermore, extreme high RH was suggested to cause stress by the induction of free radicals in the insect’s body [42]. Rivnay and Meisner [53] showed that pupae incubated at RH > 95% took longer to develop and produced adults with a shorter lifespan and with a higher frequency of barren females. In this study, larvae exposed to extreme high humidity performed notably poorer than larvae exposed to the respective low RH treatment. This is apparent in the damaged leaf area and throughout the 3 days of body length measurement (Table 4 and Figure 6). These trends show that extreme high RH causes a harmful effect on the larvae, regardless to the presence of conidia or formulation, though such RH is not likely to occur in the field, such findings shed new light on the role of humidity in pathogenic interactions of EPF.

During infection, entomopathogens change the hosts physiology to suit their lifestyle [22,56,57,58]. As EPF infect their host, they cause a reduction in the host’s energy conversion rate (i.e., the ability to convert food into body mass) due to the energetic requirements of activating the immune system [4,59]. EPF infection also affects feeding behavior, albeit in an inconsistent fashion. While many insects reduce feeding during infection of viral, bacterial and fungal entomopathogens, some insects will “self-medicate” by overeating in order to increase protein intake [4,43,56,59]. It is noteworthy that, according to the findings in this study, *S. littoralis* responds to infection by excessive eating, rather than disease-induced anorexia, as seen in the differences in damaged leaf area between conidia in aqueous solution and the control. However, probably due to the reduction in the larvae’s ability to convert food into body mass, this EPF-mediated effect is visible in damaged leaf area, but not in larval body length; this was visible, yet not significant, only when conidia were formulated in an aqueous solution, as the emulsion’s anti-feedant effect completely masked this difference.

Sub-lethal effects can be a good tool to assess the damage inflicted on the larvae, and might give more accurate criteria to the agricultural effect of a pesticide [4,49]. In this project, all tested factors influenced larval abilities to eat and develop, but not in an equal manner. It is amply clear that paraffinic oil had the most profound effect on larval growth and feeding, followed by RH differences. The presence of conidia, however, did not affect larval growth [29,30].

### 4.5. Disease Development

On the first DPI, counterintuitively, larvae exposed to conidia in aqueous solution at low RH displayed the most advanced disease, by having a significantly higher proportion of larvae with post-germinated conidia and a low proportion of non-inoculated larvae. Conidia in emulsion at high RH showed a significantly lower disease rate compared to other treatments, albeit this significance was nullified by the FWER correction used. In the initial attachment and recognition phase, conidia attach via a non-specific hydrophobic interaction [11,45]. A major determinant for conidial germination is water availability. The lowest water activity that was reported for growth was a_w_ = 0.90 (corresponding to 90% RH) [15,20,21,44]. However, Milner et al. [21] speculated that under RH lower than 90%, this requirement for high water availability is mediated by water evaporation through the insect cuticle. Likewise, on the first DPI, we do not see a reduction in the frequency of larvae with germinated conidia at low RH. This might indicate that the cuticle of first instar *S. littoralis* larvae is permeable enough to facilitate conidial germination. Similar effects on the insect cuticle can be attained by using sucrose esters and other botanical extracts, that were even found to increase permeability of the cuticle to botanic toxins [52,60]. Nonetheless, the relatively high percentage of non-inoculated larvae in the emulsion treatments, in comparison to the corresponding aqueous solution treatments, require further explanation. In all treatments on the first DPI, the proportion of larvae seen with non-germinated conidia is roughly constant (22.5–27.4%), while the proportion of non-inoculated larvae and larvae presenting post-germination stages seem to be complementary. St. Leger et al. [16] and Lazzarini et al. [15] reported that *M. anisopliae* and *B. bassiana* conidia require 4–8 h to begin germination at optimal conditions. It is therefore reasonable to assume that larvae with germinated conidia contacted conidia at least 4 h before examination, while non-inoculated larvae did not come in contact with conidia in the time before examination. This is best explained by the changes to insect behavior caused by extreme high humidity and paraffinic oil, both known to reduce insect activity [30,42]. This lack of activity is also confirmed by our sub-lethal effects test. Results show that the most advanced disease was in larvae exposed to conidia in aqueous solution at low RH, with both RH and formulation exerting minimal stress on the larvae, while the least inoculated were larvae exposed to high levels of stress caused by both high RH and paraffinic oils. 

On the second DPI, the difference was only found between formulations, regardless of RH. Emulsion formulated conidia showed a more rapid disease progression, and a lesser proportion of larvae with non-germinated conidia. These differences between formulations can be explained by the effect of oils on the insect cuticle in two fashions: disruption of the epicuticle and increasing cuticle-permeability, and a general softening of the cuticle [30]. Lazzarini et al. [15] reported that reduction in water availability prolongs the time to conidial germination. An increased water permeability of the cuticle can result in a more favorable microclimate on the insect cuticle and increased germination over time. Moreover, the softening of the cuticle can decrease its effectiveness as a physical barrier, allowing the appressoria to penetrate with greater ease.

On the third DPI, conidia formulated in emulsion at high RH showed the most advanced disease stage. Although the importance of high humidity in the germination and penetration processes is well established [15,16,22,23,44], high humidity might play a different role in pathogen–host interaction after penetration as well. After penetration, the host employs the innate immune system, Cotter et al. [56] found that diet, mostly protein intake, is a crucial modulator of immune response. This suits well with disease progression on the third DPI; as only emulsion-formulated conidia at high RH had a significantly higher median disease stage, with higher percentage of larvae showing high infection. This rapid in-host growth can be attributed to the larvae’s weakened state, as reflected by the sub-lethal effects exerted on larvae exposed to emulsion at high humidity. Besides the effects of poor diet on immunocompetence, Li et al. [42] found that humidity close to saturation significantly reduces immunocompetence and may even produce reactive oxygen species (ROS) within the hemocoel, further reducing the larva’s ability to resist infection. Boomsma et al. [61] hypothesized that generalist species, such as *M. brunneum*, will be opportunistic in nature, and require their host to be in a reduced state of physiological stress to create a successful infection. Therefore, it is possible that death through mycosis will occur in *S. littoralis* only if several stress factors are inflicted on the larva, here caused by an emulsion formulation and RH close to saturation.

## 5. Conclusions

In this project, we examined the efficacy of an emulsion formulation in several avenues: physical interaction with host and sprayed substrate, effects of humidity on mortality and infectivity, and effects on the insect’s growth and feeding. From the results obtained here, most effects of oils on EPF efficacy, seem to stem from paraffinic oils’ effects on the larvae and as a carrier rather than direct influence on conidial physiology. Emulsion formulation disperses conidia better than water with surfactants and increases the number of conidia adhered to larvae, resulting in more even and repeatable application, which explains the efficacy of ultra-low volume application of oil-formulated conidia. Furthermore, emulsion formulations induce a more rapid penetration and colonization of the host hemocoel. Interestingly, differences in mortality rates between formulations did not manifest in disease progression, as conidia in aqueous solution were able to infect larvae at the same rate as conidia in emulsion at low RH after three days, yet the corresponding mortality between these treatments was significantly higher in favor of the emulsion. The results presented in this study show that emulsion formulations can improve EPF application while improving the chances of a lethal infection by modulating in- and on-host factors, making EPF an ever more relevant substitute to conventional pesticides.

## Figures and Tables

**Figure 1 jof-07-00499-f001:**
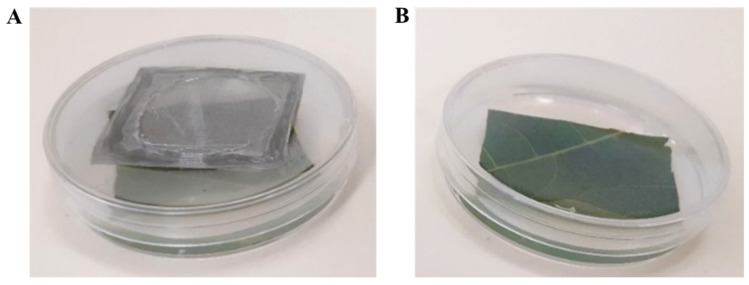
Set-up of an indirect inoculation system. Treated leaves were embedded in 2% agarose and allowed to cool and set. Plates were populated with larvae of *Spodoptera littoralis* and capped with perforated cap for low relative humidity (RH) condition (**A**) or unperforated cap for high RH condition (**B**).

**Figure 2 jof-07-00499-f002:**
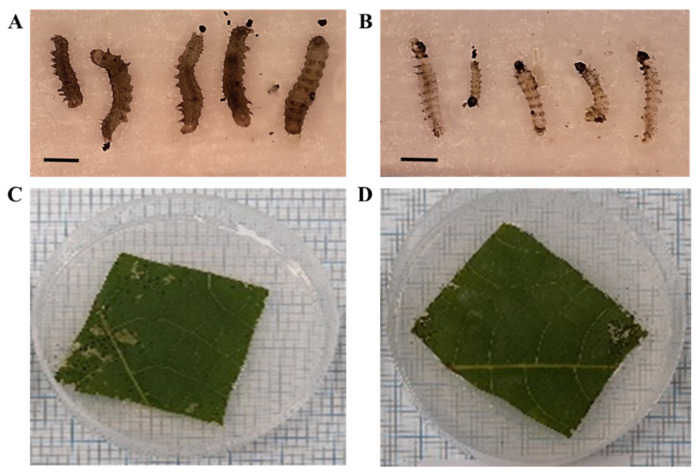
To assess larval development and larval feeding area, larvae and sacrificed plates were photographed, respectively. All pictured were taken with a background of millimeter paper (each small square is 1 mm by length and width) to set a scale for later image analysis. (**A**) Larvae exposed to aqueous solution formulated conidia after three days at low RH. (**B**) Larvae exposed to emulsion formulated conidia after three days at low RH. (**C**) Leaves treated with aqueous conidia after three days of larval feeding at low RH. (**D**) Leaves treated with Emulsion formulated conidia after three days of larval feeding at low RH.

**Figure 3 jof-07-00499-f003:**
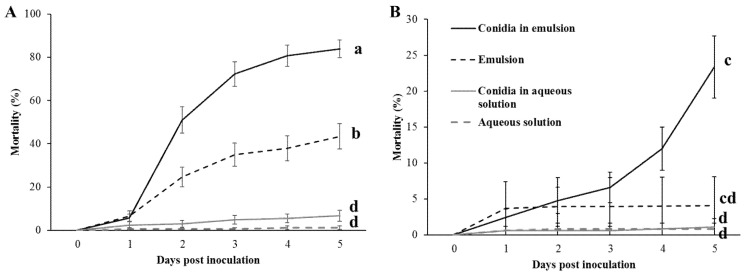
Mortality (% ± standard error (SE)) of larvae exposed to leaves treated with emulsion and aqueous solution (Triton-X100 0.01% *v*/*v*), with or without conidia, at high RH (**A**) and at low RH (**B**), over a period of 5 days. Different lower-case letters designate significant difference in fifth day post-inoculation (DPI) mortality, as was tested by Tukey’s honestly significant difference (HSD, α < 0.05).

**Figure 4 jof-07-00499-f004:**
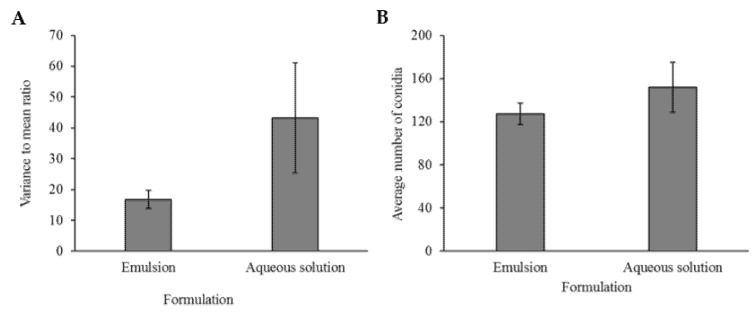
(**A**) Variance to mean ratio ±SE by treatment. If 0 < variance-to-mean ratio (VMR) < 1 distribution is regularly spaced, if VMR = 1 distribution is random, if 1 < VMR distribution is clustered. A Welch analysis of variance (ANOVA) was employed, and no significant difference was found between the distribution pattern of the conidia in the various formulations, yet a significant difference was found in the variance (Bartlett’s test: F = 19.5792, *p*-value < 0.0001). (**B**) Mean numbers of conidia ±SE counted in the various microscopic fields used to calculate VMR. A Welch ANOVA was employed, and no significant difference was found between treatments, yet a significant difference was found in variance (Bartlett’s test: F = 13.9549, *p*-value = 0.0002).

**Figure 5 jof-07-00499-f005:**
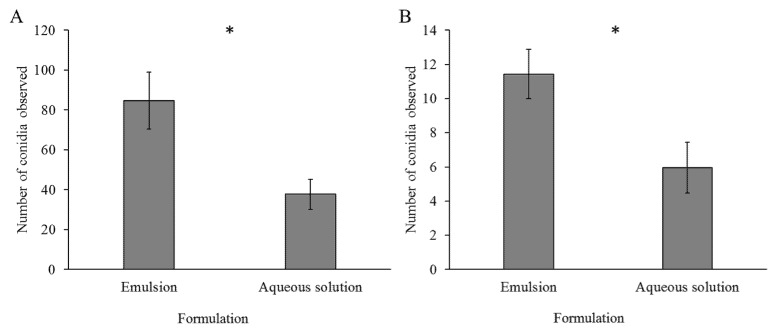
Mean number (±SE) of conidia unbound (**A**) or bound (**B**) to larvae, by formulation. To assess conidial attachment *S. littoralis* first instar larvae were exposed to leaf segments treated with conidia formulated in either aqueous solution or emulsion. After 24 h of exposure, larvae were examined under a confocal microscope to quantify the number of conidia. Larvae were not washed for acquisition or washed with tap water for adhesion. A least square analysis was employed, * indicates significant difference between formulations (FA = 4.3518, pvA = 0.0463, FB = 16.1616, pvB = 0.0002).

**Figure 6 jof-07-00499-f006:**
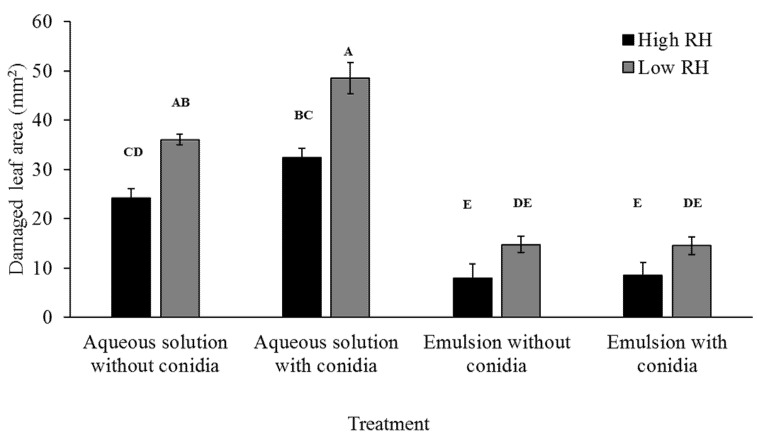
Mean leaf damage area in mm^2^ (±SE) caused by *S. littoralis* larvae exposed to different treatments and RH. The damaged area was assessed after three days of larval activity on treated leaves. The damaged area was assessed by photographing the leaves alongside millimeter paper and then analyzed using the ImageJ software. Different upper-case letters designate significant difference between experimental groups as tested via Tukey-HSD (α < 0.05).

**Figure 7 jof-07-00499-f007:**
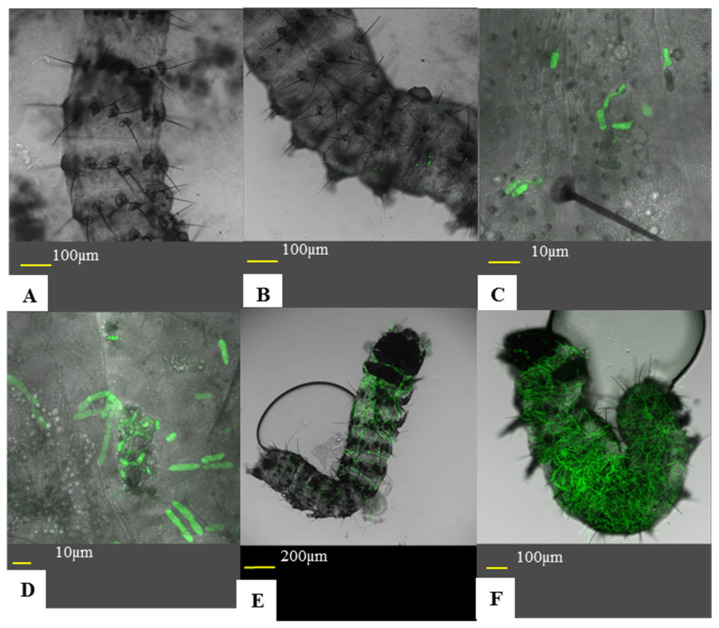
Illustrations of the various disease stages. In order to quantify disease progression, the disease caused by *M. brunneum* was sectioned into six stages: (**A**) No adhered conidia on the insect’s cuticle. (**B**) Adhered, non-germinated conidia on the cuticle. (**C**) Conidial germination and appressorium formation. (**D**) Cuticle penetration, up to 5% of hemocoel colonization. (**E**) Early infection (colonization of 5–50% of the hemocoel). (**F**) High infection (colonization of over than 50% of the hemocoel).

**Figure 8 jof-07-00499-f008:**
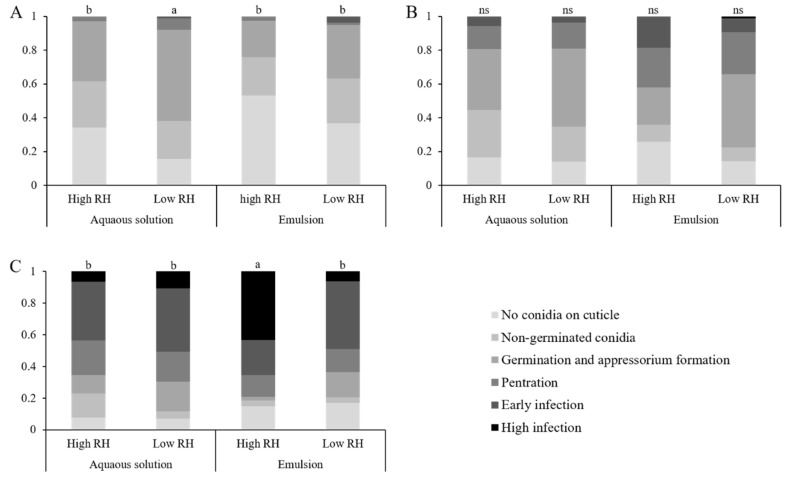
Distribution of disease stages amongst larval cohorts (for sample size see Table 1) over time. Larvae were exposed to leaves treated with aqueous solution or emulsion-formulated conidia at high or low RH. Larvae were sacrificed daily and examined under a confocal microscope. The disease stage of every larva was recorded and shown above as a percent of the total larvae examined: (**A**) one DPI (**B**) two DPI (**C**) three DPI. An each-pair Wilcoxon test was employed, different lower-case letters indicate a significant difference between treatments, ns indicates no significant difference between treatments. *p*-values were then subjected to a Holm–Bonferroni family-wise error rate (FWER) correction (α < 0.05).

**Table 1 jof-07-00499-t001:** Total sample sizes of larvae used to assess disease progression and larval development (length) ^1^.

Days Post Inoculation	Treatment	High RH		Low RH	
		Disease Progression	Larval Length	Disease Progression	Larval Length
1	Control aqueous solution		70		72
	Conidia in aqueous solution	73	75	76	74
	Control emulsion		73		68
	Conidia in emulsion	79	72	79	68
2	Control aqueous solution		73		73
	Conidia in aqueous solution	72	67	78	72
	Control emulsion		62		66
	Conidia in emulsion	81	47	76	62
3	Control aqueous solution		79		77
	Conidia in aqueous solution	78	72	85	81
	Control emulsion		32		65
	Conidia in emulsion	81	36	82	71

^1^ Larvae that were not exposed to conidia were not used for disease progression assessment.

**Table 2 jof-07-00499-t002:** Synergism assessment based on Bliss’s model for independent joint action of chemical pesticides.

	High RH ^1^	Low RH
Natural mortality (control aqueous solution)	1.31%	1.19%
Conidia induced mortality (conidia in aqueous solution)	6.79%	2.48%
Emulsion induced mortality (control emulsion)	43.59%	13.00%
Conidia in emulsion induced mortality (predicted)	48.10%	16.08%
Conidia in emulsion induced mortality (observed)	84.64% (92.109%, 77.72%) ^2^	23.57% (32.55%, 14.69%) ^2^

^1^ indicates significant difference between predicted and observed mortality from emulsion formulated conidia. ^2^ 95% confidence interval is given in brackets.

**Table 3 jof-07-00499-t003:** Statistical significance of the effect of all fixed variables and their interactions on mortality assays.

Factor	Statistic F	*p*-Value
Relative humidity	62.7329	<0.0001
Formulation	194.6273	<0.0001
Presence of conidia	19.0527	<0.0001
RH × formulation	51.205	<0.0001
RH × conidia	9.3726	0.0025
Formulation × conidia	9.9	0.0019
RH × formulation × conidia	5.1865	0.0238

**Table 4 jof-07-00499-t004:** Mean larval body length (±SE) in mm by treatment and RH.

Treatment	1 DPI		2 DPI		3 DPI	
	High RH	Low RH	High RH	Low RH	High RH	Low RH
Control aqueous solution	1.24 C (±0.02)	1.38 AB (±0.018)	2.00 AB (±0.04)	2.23 A (±0.03)	2.63 AB (±0.04)	2.66 AB (±0.03)
Conidia in aqueous solution	1.33 B (±0.02)	1.42 A (±0.01)	1.98 B (±0.03)	2.04 AB (±0.03)	2.49 BC (±0.04)	2.74 A (±0.04)
Control emulsion	1.17 DE (±0.01)	1.20 CD (±0.02)	1.43 E (±0.04)	1.67 CD (±0.056)	1.93 D (±0.08)	1.97 D (±0.07)
Conidia in emulsion	1.13 E (±0.01)	1.177 CDE (±0.02)	1.52 DE (±0.05)	1.74 C (±0.04)	2.08 D (±0.07)	2.38 C (±0.06)

Different upper-case letters designate significant difference within the same DPI, as found by Tukey-HSD (α < 0.05).

**Table 5 jof-07-00499-t005:** Statistical significance of the effect of all fixed variables and their interactions on larval body length by days post-inoculation.

Fixed Effect Factor	1 DPI		2 DPI		3 DPI	
	*p*-Value	Statistic F	*p*-Value	Statistic F	*p*-Value	Statistic F
RH	<0.0001	42.98	<0.0001	54.64	<0.0001	16.14
Formulation	<0.0001	220.36	<0.0001	494.58	<0.0001	160.53
Conidia	0.054	3.70	0.176	1.84	0.042	4.15
RH × Formulation	0.001	10.73	0.003	8.78	0.339	0.56
RH × Conidia	0.714	0.13	0.793	0.07	0.003	8.57
Conidia × Formulation	<0.0001	14.83	<0.0001	20.59	0.002	9.28
RH × Conidia × Formulation	0.126	2.44	0.301	0.58	0.835	0.04

**Table 6 jof-07-00499-t006:** Statistical significance of the effect of all fixed variables and their interactions on damaged leaf area after three days of larval activity.

Factor	Statistic F	*p*-Value
Relative humidity	51.79	<0.0001
Formulation	261.33	<0.0001
Presence of conidia	11.21	0.0011
RH × formulation	8.16	0.0051
RH × conidia	0.58	0.4445
Formulation × conidia	10.73	0.0014
RH × formulation × conidia	1.04	0.309

**Table 7 jof-07-00499-t007:** Statistical significance of various factors on disease progression, as examined by a non-parametric Wilcoxon text, by day post-inoculation.

Factors	1 DPI			2 DPI			3 DPI		
	χ2	DF	*p*-Value	χ2	DF	*p*-Value	χ2	DF	*p*-Value
Relative humidity	13.632	1	0.0002	0.429	1	0.512	3.957	1	0.046
Formulation	15.514	1	<0.0001	7.098	1	0.007	4.066	1	0.043
Relative humidity + formulation	29.332	3	<0.0001	7.674	3	0.053	19.774	3	0.0002

## Data Availability

Not applicable.

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
