# Peer review of "Not Only a Formulation: The Effects of Pickering Emulsion on the Entomopathogenic Action of Metarhizium brunneum"

_jof, 2021, doi:10.3390/jof7070499_

Round 1

Reviewer 1 Report

The work is improved but, in my opinion, the discussion is too long and the results are repeated rather than interpreted. Specific comments are included in below

-There are paragraphs that could be deleted: lines 486-492 (already noted in the results), lines 499-504 (already stated in the introduction), lines 609-615 (repeat the results), lines 332-333 (it is obvious).

-Lines 595-597 seem to contradict the results, based on figure 6. Larvae treated with aqueous solutions caused the most damage to the leaf, indicating a greater effect due to formulation rather than infection.

-Lines 626-667, the discussion of this part would be more dynamic if the interpretation were more global and examined the effect on development in the period studied for each solution.

-Lines 653-659: This paragraph is difficult to understand and it is not clear what the interpretation of these results is.

-And finally, regarding figure 7, this figure could be deleted, as it does not provide the detail of what is to be represented. Moreover, the figures should be mentioned in order in the text, and this figure is the first to be mentioned.

Author Response

Dear reviewers, we like to thank you for taking a second time review this work. We have done our best to take notice of your notes, while keeping this work accessible for readers and true to our original intent.

The work is improved but, in my opinion, the discussion is too long and the results are repeated rather than interpreted. Specific comments are included in below

-There are paragraphs that could be deleted: lines 486-492 (already noted in the results), lines 499-504 (already stated in the introduction), lines 609-615 (repeat the results), lines 332-333 (it is obvious).

For lines 332-333, though you are right, and the method used is the most probable, as well as fully described in the materials and methods, we feel it would be easier for readers to understand the text if it would be shortly described for context. We find lines 486-492 and lines 499-501 imperative to the narrative displayed. While lines 486-492 corroborate our results with those of past studies, lines 499-501 present a prediction to what will happen in longer trials. As for lines 609-611 were dropped out.

-Lines 595-597 seem to contradict the results, based on figure 6. Larvae treated with aqueous solutions caused the most damage to the leaf, indicating a greater effect due to formulation rather than infection.

Further explanations were set to fully explain the phenomenon. This increase in feeding (damaged leaf area) can be seen if comparing feeding within the same formulation and RH category.

-Lines 626-667, the discussion of this part would be more dynamic if the interpretation were more global and examined the effect on development in the period studied for each solution.

As our examinations were lethal to the larvae, no repeated measures were possible. Therefore, we thought it best to find differences between formulations and RH levels, describe and explain said differences, as it would be more representative to our work and would reduce assumptions that might not be true.

-Lines 653-659: This paragraph is difficult to understand, and it is not clear what the interpretation of these results is.

Phrasing was changed as to better focus the reader to our intent.

-And finally, regarding figure 7, this figure could be deleted, as it does not provide the detail of what is to be represented. Moreover, the figures should be mentioned in order in the text, and this figure is the first to be mentioned.

Though this figure can be removed, we believe it will make our method of quantifying disease stages less understandable. Furthermore, we felt that putting figures 7 and 8 together would make it easier for the reader to interpret figure 8 and visualize the process of infection. We do understand that mentioning the seventh figure firstly is out of general custom and form, yet we are positive this is best for the accessibility of both results and methods.

Reviewer 2 Report

the authors have done an interesting piece of work in the use of  EPF application in feild. the objectives are clearly defined and the work carried out is systematic and obtained results are meaningful.

Author Response

Dear reviewers, we like to thank you for taking a second time review this work. We have done our best to take notice of your notes, while keeping this work accessible for readers and true to our original intent.

This manuscript is a resubmission of an earlier submission. The following is a list of the peer review reports and author responses from that submission.

Round 1

Reviewer 1 Report

This study analyzes the effect of Pickering emulsion on the infection process of the fungus Metarhizium brunneum on Spodoptera litoralis larvae. The work is scientifically consistent, provides a detailed methodology and results clearly described. But there are some aspects that could be improved.

Additional comments are included in below

The introduction section should be thoroughly revised, it is long and there are sentences that say practically the same, these sentences could be unified and simplified, as an example: between lines 81-91. Also, it would need more detailed information in the context of the research, for example, it is hardly said of other works in which the activity of this oil on fungi or other organisms have been studied nor of the effect of spores of this funus on insects, in particular Spodoptera. And lastly of this section, the objectives of the work described at the end of the introduction should be reconsidered and rewritten.

In methodology section, the insect rearing is not enough detailed

The results section is very dense, making it heavy to read, because it is described all data, whether significant or not, relevant or not for the objective of the study. They should be simplified, explaining the most important data, since the rest of the information is provided in the tables and figures. By other hand, the way of providing the data can lead to confusion, for examples: lines 357-359, 367-369, and so on: where firstly a result based on the absolute value is provided, but which lacks validity as it is not significant.

The discussion section should be entered in the results section since some of them are repeated in each of the sub-sections, as they are discussed with the same structure in which they were described, without offering a global and integrating overview of them.

The conclusions section should be modified. This section should only include a synthesis of the results that led to achieving the objective proposed in the research. It should not include notes of the introduction or discussion.

Respect to other details are:

- The sentences in lines 393-396 and 506-510 are difficult to understand.

- The discussion offered between lines 697-707 should be revised to improve its contextualization with the results obtained.

- The tables should be mentioned in order in the text, table 3 is named before table 2; the same happens with the abbreviation of water distilled (WP), which appears for the first time on line 144 and its abbreviation is indicated on line 153.

- The figure 1 should be eliminated or modified, as it is not more informative than the text

- The legend in Figure 5 should be revised and simplified.

- Figure 8 should be labeled with the most important elements.

Author Response

To all it may concern, thank you for making the time to review and comment on this work. We take your comments with the utmost care and attention and do our best to incorporate them into this study.

Reviewer 1

Comments and Suggestions for Authors

This study analyzes the effect of Pickering emulsion on the infection process of the fungus Metarhizium brunneum on Spodoptera litorallis larvae. The work is scientifically consistent, provides a detailed methodology and results clearly described. But there are some aspects that could be improved.

Additional comments are included in below

The introduction section should be thoroughly revised, it is long and there are sentences that say practically the same, these sentences could be unified and simplified, as an example: between lines 81-91. Also, it would need more detailed information in the context of the research, for example, it is hardly said of other works in which the activity of this oil on fungi or other organisms have been studied nor of the effect of spores of this fungus on insects, in particular Spodoptera. And lastly of this section, the objectives of the work described at the end of the introduction should be reconsidered and rewritten.

Thank you for the keen observation. Dualities in text were deleted and revised to streamline the text and the research objectives were revised. In consideration of the various topics of the introduction, we have done our best effort to include only relevant subjects and to minimize repetition in the discussion section. To that extent, efficacy of other organisms and formulations, which were deemed most relevant in the discussion section, were not included in the introduction.

In methodology section, the insect rearing is not enough detailed

This comment is pertinent, and information as to the age of the insect colony and the place where the first insects were collected is common practice. However, as this colony was not established by our lab, we could not attain this knowledge.

The results section is very dense, making it heavy to read, because it is described all data, whether significant or not, relevant or not for the objective of the study. They should be simplified, explaining the most important data, since the rest of the information is provided in the tables and figures. By other hand, the way of providing the data can lead to confusion, for examples: lines 357-359, 367-369, and so on: where firstly a result based on the absolute value is provided, but which lacks validity as it is not significant.

All results were laid bare for the reader to see, as to allow complete transparency of the inference process. However, non-significant elements and raw data are shortened here as to not over-burden the reader.

The discussion section should be entered in the results section since some of them are repeated in each of the sub-sections, as they are discussed with the same structure in which they were described, without offering a global and integrating overview of them.

As every avenue of research poses a potential improvement in EPF application, we felt that every experiment conducted deserved its own section of the discussion, as some novel formulations may only affect certain, but not all, interactions.

The conclusions section should be modified. This section should only include a synthesis of the results that led to achieving the objective proposed in the research. It should not include notes of the introduction or discussion.

Conclusion section was modified as suggested

Respect to other details are:

- The sentences in lines 393-396 and 506-510 are difficult to understand.

Sentences were clarified as much as possible, without losing their original intent

- The discussion offered between lines 697-707 should be revised to improve its contextualization with the results obtained.

Though this study did not include examination of immunocompetence of the host via biochemical or molecular tools, explanations regarding these mechanisms, stemming from the literature, explained the differences in disease progression best, to our opinion.

- The tables should be mentioned in order in the text, table 3 is named before table 2; the same happens with the abbreviation of water distilled (WP), which appears for the first time on line 144 and its abbreviation is indicated on line 153.

Tables were rearranged and abbreviations explained where they first appear.

- The figure 1 should be eliminated or modified, as it is not more informative than the text

Figure 1 was removed, and the numbering of all other figure revised.

- The legend in Figure 5 should be revised and simplified.

Figures 5A and 5B (representing photos) were removed, as to simplify both the figure and the legend.

- Figure 8 should be labeled with the most important elements.

Though figure 8 is rather cumbersome, it is paramount to demonstrate the various infection statuses. To our opinion, exclusion of parts of the figure will make the methodology the disease progression less clear.

Reviewer 2 Report

The paper describes the evaluation and assessment of oil and water formulations of Metarhizium against neonate Spodoptera littoralis in the laboratory.

The article is excessively long; from my point of view: 23 pages.
The authors should do their best to reduce this size.

I consider that Figure 1 is not necessary; although it is interesting and presents good information, but that information is already available in other sources, and it is more appropriate for a book or a review article.

In general, all the text of the article should be carefully reviewed to reduce the size of the article.

The submitted PDF file includes many comments and marks in the text.
These observations should be reviewed.

The literature section consulted requires a review of the references. Some references have capital letters that should not appear and the names of organisms are not italicized or underlined which is a universal biological convention.
There are some grammatical errors and inappropriate words.

Author Response

To all it may concern, thank you for making the time to review and comment on this work. We take your comments with the utmost care and attention and do our best to incorporate them into this study.

Reviewer 2

Comments and Suggestions for Authors

The paper describes the evaluation and assessment of oil and water formulations of Metarhizium against neonate Spodoptera littoralis in the laboratory.

The article is excessively long; from my point of view: 23 pages.
The authors should do their best to reduce this size.

The article was shortened in all parts.

I consider that Figure 1 is not necessary; although it is interesting and presents good information, but that information is already available in other sources, and it is more appropriate for a book or a review article.

Figure 1 was removed, and the numbering of all other figure revised.

In general, all the text of the article should be carefully reviewed to reduce the size of the article.

The article was shortened in all parts.

The submitted PDF file includes many comments and marks in the text.
These observations should be reviewed.

Comments were individually addressed in the body of the text and were incorporated as much as possible.

The literature section consulted requires a review of the references. Some references have capital letters that should not appear and the names of organisms are not italicized or underlined which is a universal biological convention.

References were revised according to the journal’s format.

There are some grammatical errors and inappropriate words.

Highlighted parts were thoroughly reexamined, and revised as necessary and if revision did not affect the intention of the text.

Comment made on figure 7 –

Such levels of colonization are observable after three days, and in rare cases, even after two days. Though such cases would, indeed, be rare in the field, as the experiments that were preformed under conditions optimal for fungal growth.